# FacialPulse: An Efficient RNN-based Depression Detection via Temporal Facial Landmarks

## ABSTRACT

Depression is a prevalent mental health disorder that significantly impacts individuals' lives and well-being. Early detection and intervention are crucial for effective treatment and management of depression. Recently, there are many end-to-end deep learning methods leveraging the facial expression features for automatic depression detection. However, most current methods overlook the temporal dynamics of facial expressions. Although very recent 3DCNN methods remedy this gap, they introduce more computational cost due to the selection of CNN-based backbones and redundant facial features. To address the above limitations, by considering the timing correlation of facial expressions, we propose a novel framework called **FacialPulse**, which recognizes depression with high accuracy and speed. By harnessing the bidirectional nature and proficiently addressing long-term dependencies, the Facial Motion Modeling Module (FMMM) is designed in **FacialPulse** to fully capture temporal features. Since the proposed FMMM has parallel processing capabilities and has the gate mechanism to mitigate gradient vanishing, this module can also significantly boost the training speed. Besides, to effectively use facial landmarks to replace original images to decrease information redundancy, a Facial Landmark Calibration Module (FLCM) is designed to eliminate facial landmark errors to further improve recognition accuracy. Extensive experiments on the AVEC2014 dataset and MMDA dataset (a depression dataset) demonstrate the superiority of **FacialPulse** on recognition accuracy and speed, with the average MAE (Mean Absolute Error) decreased by 22%, and the recognition speed increased by 100% compared to state-of-the-art baselines.

## CCS CONCEPTS

• **Applied computing → Life and medical sciences**.

## KEYWORDS

Depression detection, Temporal facial landmarks

## 1 INTRODUCTION

Depression is a common mental health problem. According to the World Health Organization, over 264 million people worldwide were clinically diagnosed with depression in 2020, leading to severe consequences such as addiction, impulsive behavior, and suicide. Therefore, early detection plays a crucial role in significantly

mitigating the harm caused by depression. Due to the scarcity of healthcare personnel, the exploration of automatic detection of depression gained attention in past years. In particular, human faces are acknowledged as a primary communication channel and a pivotal conduit for conveying crucial information about mental states, intentions, and personality traits. Past psychological research emphasized the reliability of non-verbal facial behaviors as indicators of depression [28]. Motivated by this, this paper aims to investigate the potential of recognizing facial emotion for early depression detection.

Latest advancements in computer vision contribute to the automatic recognition of human facial behaviors [4, 21, 31], which facilitates automated analysis of depression from facial videos [7, 9, 14, 33]. However, there are three main limitations of existing methods: 1) *Overlooking temporal facial characteristics.* Individuals with depression exhibit fewer spontaneous facial expressions of emotion compared to healthy individuals, which indicates unique temporal features are contained in the facial expressions of depressed patients. Extensive experiments demonstrated significant improvements in recognition accuracy with Convolutional Neural Networks (CNNs) over conventional methods [7, 33]. However, these CNN-based methods treat a video as a collection of static images, focusing on spatial features while inevitably overlooking temporal characteristics and the dynamic nature of facial expressions. 2) *Complex model architectures induce more computational cost.* To comprehensively capture both temporal and spatial characteristics, CNN-RNN and 3DCNN methods [9, 14] emerged as preferred choices. However, these detection methods heavily rely on complex models or data-enhanced techniques, which require longer calculation time and higher costs. 3) *Rely on redundant raw facial features.* Traditional approaches mostly relied on raw images as input. However, the use of raw images as input inevitably causes information redundancy. The main reason is that these original images may contain a significant amount of task-irrelevant information, such as background and lighting conditions, which necessitates the model to handle a surplus of redundant data.

To address the above limitations, we propose an efficient framework named **FacialPulse**, which contains the two primary modules: the Facial Motion Modeling Module (FMMM) and the Facial Landmark Calibration Module (FLCM). The motivation and introduction of these two modules are provided as follows:

- **Modeling Facial Motion Based on Temporal Sequences:** Each emotion manifests a unique temporal pattern, and the temporal modeling approach offers a novel perspective for facial expression recognition. Fig. 1 shows the motion curves of both depressed people and normal people in the same task. The shown face motions are AU12 and AU15. AU12 and AU15 represent the upward and downward movement of the mouth, respectively. It can be clearly observed that the variation curves of depressed people appear smoother than those of normal people. In contrast

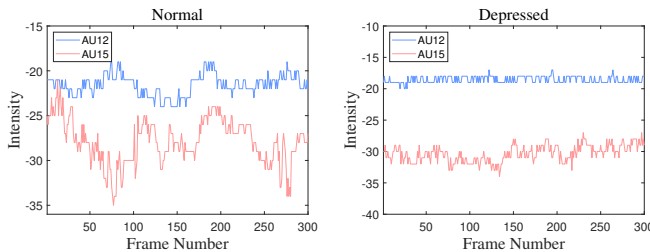

Figure 1: Change intensity of facial action units with depressed patients and normal people in the same task scenario. The vertical axis denotes the magnitude of these variations, while the horizontal axis tracks the progression of video frames.

to other mental disorders, since the facial motion changes of depression are not obvious, depression detection requires prolonged and continuous monitoring of changes in facial expressions. To better capture the characteristics of depression, by using the Bidirectional Gated Recurrent Unit (BiGRU) as the backbone, we propose a module that harnesses the bidirectional nature and addresses long-term dependencies for temporal modeling. To capture evolving patterns and characteristics more effectively, this module emphasizes the temporal sequence and contextuality of facial features. Besides, since incorporating parameter sharing and temporal dependencies, this module provides a significant advantage in training speed.

- **Facial Landmark Calibration Module (FLCM):** Facial landmarks are a set of points outlining the contours of distinctive facial features and are sufficient for describing geometric information. Thus, instead of using the original raw image as input, we choose facial landmarks as input to detect depression with less information redundancy since facial landmarks contain key points of facial information while eliminating the impact of irrelevant areas on recognition. Although previous research has demonstrated the improvement effect of facial landmarks in facial emotion recognition [29], facial landmarks are rarely emphasized in depression detection. Furthermore, existing approaches do not take into account the accumulative errors in landmark detection. To ensure the accuracy and precision of landmark detection, we further introduce a novel landmark calibration module. By minimizing jittering, this module enhances the recognition capability of landmarks, which significantly facilitates the reliable integration of landmarks and deep temporal features.

In a nutshell, by considering the distinctive temporal characteristics of facial expressions in various depressed individuals, we further combine both preceding and subsequent contextual information to analyze comprehensive temporal information. Besides, to reduce the redundancy of input information, we employ facial landmarks as input to detect depression. Furthermore, to ensure the accuracy of the landmarks and remove the accumulative errors, we propose a novel calibration module by minimizing jittering. The introduction and calibration of landmarks significantly improve the reliability of the captured temporal features.

We evaluate **FacialPulse** on two datasets (i.e., AVEC2014 [34] and MMDA [17]), which demonstrate that **FacialPulse** outperforms the baseline methods by a large margin and decreases the training time (including preprocessing time) by 2×. Overall, the main contributions of this paper can be summarized as follows:

- By using the BiGRU as the backbone, a facial motion modeling module (FMMM) is proposed to better capture the characteristics of depression. This module harnesses the bidirectional nature, addresses long-term dependencies for temporal modeling, and emphasizes the temporal sequence and contextuality of facial features, which significantly improves recognition accuracy.
- To ensure the accuracy of the landmarks and remove the accumulative errors, we propose a novel calibration module (FLCM) by minimizing jittering. The calibration of landmarks further improves the captured temporal feature reliability.
- Extensive experiments on various datasets demonstrate the superiority of FacialPulse on recognition accuracy and speed, with the average MAE (Mean Absolute Error) decreased by 22%, and the recognition speed increased by 2× compared to state-of-the-art baselines.

## 2 RELATED WORK

In this section, we first discuss the input difference related to state-of-the-art (SOTA) facial expression recognition-based depression detection methods in (Sec. 2.1). Then, the differences in network frameworks used by different SOTA methods are further discussed in (Sec. 2.2). By showing the differences in network inputs and the structure of related SOTA methods, the shortcomings and differences of existing methods are clearly highlighted.

### 2.1 Facial Landmarks Detection

Emotion recognition [20, 26] heavily relies on facial feature detection [22, 32] and it is extremely necessary to extract effective facial features. Facial landmarks, as one of the crucial facial features in various computer vision tasks [27, 36], play a pivotal role in capturing both spatial and temporal information related to facial expressions [23].

Classical parametric methods, e.g., Active Appearance Models [30], Constrained Local Models, Supervised Descent Variant Method, and Cascade Regression Algorithm [12], can effectively detect facial landmarks. Due to the user-friendly interfaces and high detection speeds, these parametric methods are widely employed and integrated into open-source image processing libraries.

Recently, deep learning models, e.g., cascade CNNs, Convolutional Pose Machines, and Constrained Local Models, have emerged in computer vision and can extract facial landmarks with high accuracy. Similarly, since the extracted landmark can significantly boost the recognition speed, these deep learning-based facial landmark extraction methods are widely integrated into open-source toolkits, like OpenFace.

Although using accurate facial landmarks for face normalization significantly improves recognition accuracy, the low quality of landmark detection directly downgrades the final system performance. Many studies [2] emphasize the significance of precision in detected landmarks. Furthermore, the intrinsic jitter noises of facial

**Figure 2: Illustration of the overall pipeline of FacialPulse, which contains two primary modules: (a) Facial Landmark Calibration Module and (b) Facial Motion Modeling Module. The input is a video and the output is the subject's BDI-II questionnaire score.**

landmarks inevitably interfere with its temporal features. However, existing SOTA methods do not effectively calibrate the various errors of landmarks, which further makes it difficult to promote the method. To overcome these issues, a calibration module is proposed in this paper (called FLCM) to eliminate the accumulative errors for higher accuracy of landmark detection.

## 2.2 Facial-Based Automatic Depression Recognition

Psychological studies [8][18] indicate that the variations in facial expressions can serve as predictive indicators of individual depression severity. Consequently, numerous researchers endeavor to establish the mapping between facial features and depression scores via machine learning techniques.

Initially, hand-crafted methods generally utilize specific feature descriptors to represent depression, where Edge Orientation Histogram (EOH) and Local Binary Pattern (LBP) are used as spatial features to encode images. For example, He *et al*. [15] proposed the MRLBP-TOP framework to capture spatial information of facial microstructure in video segments. Subsequently, a local pattern LSOGCP [25] was proposed to further extract detailed facial texture. However, the hand-crafted features used in the aforementioned works heavily rely on experience and expertise, which implies that the essential information related to depression may be lost when manually extracting features.

To overcome this problem, researchers are inclined to detect depression based on deep learning architecture, especially CNN-based models. Specifically, He *et al*. [13] parted the face into 24 small blocks and further adopted attention mechanisms and an aggregation method to enhance spatial significance. Meanwhile, Melo *et al*. [9] proposed that inserting maximizing and differentiation blocks into 2D-CNNs to capture facial changes can improve recognition accuracy.

To further capture spatiotemporal features to detect depression, various SOTA methods employed 3D-CNNs to encode temporal information. For instance, Zhou *et al*. [38] developed a strategy based on 3D-CNN that combines label distribution and metric learning

to enhance the representation capability for spatiotemporal information. C3D technology was employed in [7] to extract spatiotemporal features to enhance depression-related information through attention blocks. This operation effectively reduced noise and summarized video-level depression information. Similarly, He *et al*. [14] proposed a 3D CNN framework equipped with a spatiotemporal feature aggregation module to accurately characterize depression cues in video segments.

Although the above methods achieve satisfactory performance by extracting facial depression information with CNNs, most of them require high time complexity. Furthermore, these methods overlook the continuity of facial expressions in depressed individuals, which results in the limitation of detecting temporal features of depression. To effectively solve this problem and to better capture correlations in consecutive facial expressions, by focusing on the temporal sequence of facial expressions, we devise a target FMMM to capture the accurate depression characteristics.

## 3 METHODS

### 3.1 Overview

The workflow of the proposed depression detection framework **FacialPulse** is illustrated in Fig. 2, which is composed of two modules: Facial Landmark Calibration Module (FLCM) in Sec. 3.2 and Facial Motion Modeling Module (FMMM) in Sec. 3.3. In particular, the FLCM is used for the meticulous calibration of facial landmarks to eliminate accumulative errors while the FMMM is employed to cope with long-term dependencies for temporal modeling and emphasize the temporal sequence and contextuality via the bidirectional nature of BiGRU.

### 3.2 Facial Landmark Calibration Module

To effectively extract facial Landmarks from original images, we first conduct face detection on each video frame to estimate the facial bounding box and preserve the region of interest that contains the face. Then, based on the processed facial image, 68 facial landmarks are further extracted to outline the facial contour. Finally,

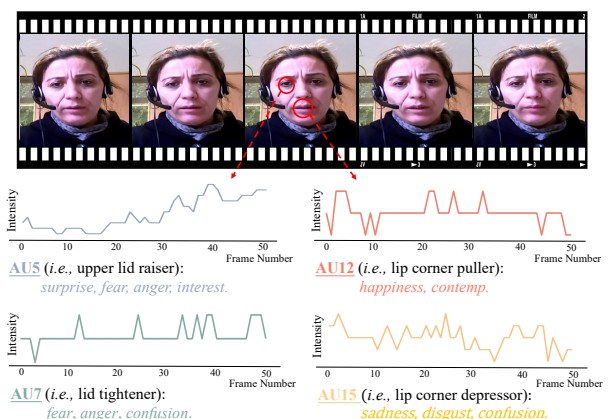

**Figure 3: Illustration on the movements of facial landmarks during an expression that appears stable. An Action Unit (AU) is calculated by two specific landmarks, representing different action areas. For example, AU5 depends on the 22nd and the 23rd Landmarks, while AU12 and AU15 correspond to those related to the mouth.**

an affine transformation [37] is employed to achieve point-to-point alignment [11] and localization.

Given the fact that landmark detection is essential for capturing facial features, how to guarantee the accuracy of facial landmarks is a critical issue. Fig. 3 shows the movements of different facial landmark units during an expression that appears stable. AU5 and AU7 denote the units near the eye area, while AU12 and AU15 express the units near the mouth area. We can observe that despite seemingly stable facial expressions, there is still a discernible fluctuation in facial landmarks, which significantly disrupts temporal consistency. This phenomenon clearly explains the importance of effectively detecting facial landmarks.

Facial movements tend to be smaller during depression expressions. Unfortunately, facial landmark detection noise has a greater negative impact in scenarios with minor facial movements. Hence, it is significant to obtain a more accurate sequence of facial landmarks when detecting depression. To solve this problem, we design the Facial Landmark Calibration Module to mitigate the impacts of abnormal fluctuations for further improving the detection accuracy of facial landmarks. FLCM is composed of motion landmark prediction and landmark error filtering, which we will introduce in detail below.

*3.2.1 Motion Landmark Predition.* During dynamic changes in facial expressions, the position of landmark pixels should remain almost the same during small periods. However, there may be some jitter in the actual detected facial landmarks. Landmarks with large jitter will cause significant errors in the detection results. Therefore, we use the optical flow algorithm to predict the facial landmark position of the current frame to provide a reference for the detection results of the next frame. Then, we compare the predicted facial landmark with the currently detected facial landmark. Points with a large difference between the detected value and the predicted value indicate that there is a larger jitter and these points will be discarded.

In particular, the sparse optical flow can selectively track a subset of points in the image rather than track all points. Considering the proposed motion estimation is based on facial key point sequences, we adopt the sparse optical flow to predict motion landmarks. Due to the reduction of tracking points, the use of sparse optical flow further improves the training speed.

Assuming that the pixel coordinates $I(x, y, t)$ in the initial frame denote the value of the pixel $I(x, y)$ at time $t$, and the pixel moves $(d_x, d_y)$ after a time interval $d_t$. Since the pixel is usually stable over a short period and its intensity remains constant, this process can be formulated as:

$$I(x, y, t) = I(x + d_x, y + d_y, t + d_t). \tag{1}$$

Assuming the motion is negligible over a short period, Taylor's formula can be employed to express this relationship. Thus, the Eq. 1 can be reformulated as:

$$\frac{\partial I}{\partial x}\frac{\partial x}{\partial t} + \frac{\partial I}{\partial y}\frac{\partial y}{\partial t} + \frac{\partial I}{\partial t} = 0, \tag{2}$$

where $\frac{\partial I}{\partial x}$ and $\frac{\partial I}{\partial y}$ denotes the gradient of pixel $I$ in the horizontal direction ($x$ direction) and vertical direction ($y$ direction), respectively. For simplicity, we represent $\frac{\partial I}{\partial x}$ and $\frac{\partial I}{\partial y}$ as $I_x$ and $I_y$. Besides, the coordinate change velocity parameters $\frac{\partial x}{\partial t}$ and $\frac{\partial y}{\partial t}$ are denoted as $u$ and $v$, respectively. Hence, the Eq.2 can be simplified as follows:

$$I_x u + I_y v + I_t = 0, \tag{3}$$

where $u$ and $v$, as two parameters of the optical flow field, numerically describe changes in pixel positions between adjacent frames. These two parameters can directly represent the motion of objects or scenes in the image.

There are significant challenges in calculating the parameter values of $u$ and $v$. When we use one single pixel to calculate the corresponding parameter values, there are two unknown parameters that cannot be effectively solved from one motion equation. Considering that adjacent points within the same exhibit similar motions, we first choose several points (the chosen point number denoted as $\gamma$) in an adjacent block matrix $n \times n$ to replace a single pixel to achieve a target that uses multiple equations to solve the goal of two unknown parameters. Then, we employ the Lucas-Kanade (LK) algorithm [40] to calculate the values of $u$ and $v$, i.e.,

$$I_{x1} u + I_{y1} v = -I_{t1},$$
$$I_{x2} u + I_{y2} v = -I_{t2},$$
$$\vdots$$
$$I_{x\gamma} u + I_{y\gamma} v = -I_{t\gamma}. \tag{4}$$

Due to the fact that the least squares algorithm has a small error in the fitting process, we employ this algorithm to solve the above motion equations. Therefore, Eq. 4 can be rewritten as matrix form:

$$\begin{bmatrix} I_{x1} & I_{y1} \\ I_{x2} & I_{y2} \\ \vdots & \vdots \\ I_{x\gamma} & I_{y\gamma} \end{bmatrix} \begin{bmatrix} u \\ v \end{bmatrix} = \begin{bmatrix} -I_{t1} \\ -I_{t2} \\ \vdots \\ -I_{t\gamma} \end{bmatrix}. \tag{5}$$

In this way, the fitting process of parameters $u$ and $v$ can be expressed by the following equation:

$$
\begin{bmatrix} u \\ v \end{bmatrix} = \begin{bmatrix} \sum_{i=1}^{Y} I_{xi}^2 & \sum_{i=1}^{Y} I_{xi}I_{yi} \\ \sum_{i=1}^{Y} I_{xi}I_{yi} & \sum_{i=1}^{Y} I_{yi}^2 \end{bmatrix}^{-1} \begin{bmatrix} -\sum_{i=1}^{Y} I_{xi}I_{ti} \\ -\sum_{i=1}^{Y} I_{yi}I_{ti} \end{bmatrix}. \quad (6)
$$

Since the size of the adjacent block matrix $n \times n$ is a fixed number, the LK algorithm that solves the values of the coordinate change velocity parameters $\frac{\partial x}{\partial t}$ and $\frac{\partial y}{\partial t}$ may not be adaptive to pixel motions at different scales.

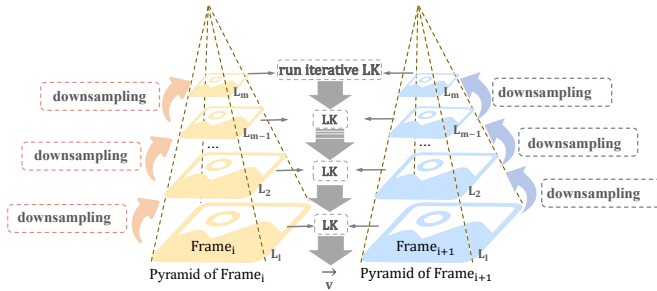

**Figure 4: PLK estimates the optical flow of feature points by employing the LK algorithm on individual layers of the image pyramid. It iteratively refines the position and optical flow vector of feature points across layers, enhancing both the accuracy and stability of estimation.**

To address this challenge, we employ the Pyramid Lucas-Kanade (PLK) architecture to adaptively capture motion information of pixels at different scales. Fig. 4 illustrates the workflow of the PLK algorithm in our depression detection task. Specifically, to reduce unnecessary calculations, we first construct a Gaussian pyramid of input images, where each level represents a different scale. Then, for each landmark pixel, iterating from the roughest granular scale, the optical flow estimation is performed at the top level. Subsequently, the estimated flow is propagated downward through the pyramid. For each level of the pyramid, to generate the corresponding pixels in the current layer, the pixels in the previous layer are aligned to adapt to the current layer resolution. Furthermore, the LK algorithm is employed to process current layer pixels to estimate the motion information. It is worth noting that this process continues until reaching the bottom level. Finally, the estimated flows of all levels are combined to obtain the final motion estimation.

To further minimize errors caused by the calculation order, we introduce a two-way error denoise mechanism. Specifically, we first compare the difference in pixel motion information calculated using the PLK algorithm from front to back and from back to front. Then, we use the threshold method [3] to process the difference between these two pixel motion information. According to the threshold judgment, we further decide whether to use the landmark pixel for depression detection. Fig. 5 depicts the selection process of facial landmarks. We can observe that the landmark pixels with significant positional differences calculated by the two-way error denoise mechanism are discarded for more accurate landmark prediction. On the contrary, the landmark pixels with tiny positional differences are retained to detect depression.

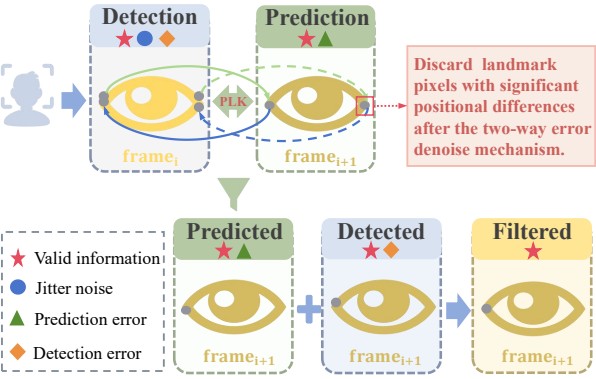

**Figure 5: The facial landmark calibration process.**

*3.2.2 Landmark Error Filtering.* Fig. 5 depicts the total workflow of the proposed calibration algorithm. During the entire calibration process, three types of errors are effectively calibrated, namely, jitter noise (denoted by the blue circle), flow prediction error (denoted by the green triangle), and detection error (denoted by the orange rhombus). The jitter noise is caused by the actual detected point jitter. Although different facial landmark points are effectively selected by positional differences (jitter noise elimination), there are still a lot of flow prediction errors caused by the LK algorithm in the prediction process. This is because the LK algorithm is a fitting algorithm and cannot obtain accurate analytical points, there are errors in the fitting process. These errors are actually flow prediction errors. Furthermore, due to changes in facial expressions and poor image quality, there are also errors in the landmark detection process, and these errors are called detection errors. To compensate for the inaccuracy and limitation in these two data sources (flow prediction results and detection results), we fuse the predicted values from the previous frame and the detected values from the current frame to obtain more reliable and complete facial motion information.

Since Kalman filtering proves effective in calibrating bimodal correlation errors due to incorporating prior information into state estimation [10], we use Kalman filtering to fuse the flow prediction results and detection results to comprehensively obtain more accurate landmark point positions.

## 3.3 Facial Motion Modeling Module

Since the facial expressions of depressed individuals have unique temporal characteristics, we utilize temporal features for depression detection. Furthermore, we verified that facial landmarks can reflect fine-grained facial fluctuations even in subtle expression changes, and facial landmarks can accurately represent the feature information of facial expressions. Depressed individuals have more subtle changes in facial expressions than other mental illnesses. Therefore, we simultaneously model facial absolute positional information and relative change information in individuals with depression. We divide a video into multiple time windows and extract feature vectors in each time window to represent the characteristics of

facial expressions. Previously we have obtained the calibrated facial landmarks, given the calibrated landmark point $U = [x, y]^T$, the first type of feature vector $\boldsymbol{a_i}$ which represents facial absolute positional information, derived from landmarks $[U_i^1, ..., U_i^{68}]^T$, is generated as follows:

$$a_i = [x_i^1, y_i^1, ..., x_i^{68}, y_i^{68}]. \tag{7}$$

Then, the second type of feature vector $\boldsymbol{b_i}$ which represents facial relative change information can calculated by:

$$\begin{aligned} b_i &= a_{i+1} - a_i \\ &= [x_{i+1}^1 - x_i^1, y_{i+1}^1 - y_i^1, ..., x_{i+1}^{68} - x_i^{68}, y_{i+1}^{68} - y_i^{68}]. \end{aligned} \tag{8}$$

These feature vectors form two feature vector sequences, which represent the temporal feature changes of the entire facial expression. Based on the above process, we obtain two feature vector sequences:

$$A = [a_1, ..., a_n]^T, \tag{9}$$

$$B = [b_1, ..., b_n]^T. \tag{10}$$

Since facial expressions are dynamic and temporally dependent, and the expressions of individuals with depression may suddenly change in a short period and exist for a long period, temporal features are thus particularly critical in detecting depression. Considering that combining forward and reverse information flows, which can more comprehensively capture the information in sequence data and mitigate information loss, Bidirectional Gated Recurrent Unit (BiGRU) is chosen as the backbone of our network to accurately capture the temporal features of depression expressions. Empowered by BiGRU, the proposed module can take into account both past and future information to better understand the facial expression context and its corresponding evolution.

Specifically, we employ two BiGRU networks to encode these sequences separately. The first BiGRU ($r_1$) models facial motion patterns on sequence $A$. Its bidirectional recurrent structure is profitable for mining temporal characteristics in landmark motion, which effectively focuses on dynamic variations in landmarks across consecutive frames and precisely extracts temporal information related to depression. Then, the second BiGRU ($r_2$) processes landmark motion speed patterns on sequence $B$. By capturing temporal features of landmark differences, this network can identify subtle facial motion changes in a brief period, which further upgrades sensitivity in detecting emotional fluctuations. Additionally, since the gating mechanism of BiGRU can learn and remember patterns in sequence data more effectively, it can also converge faster than traditional methods and accelerate the training process.

The fully connected layers are employed after the output of each BiGRU, which maps the representations to the depression detection level, respectively. The outputs of both streams are averaged to obtain the final depression detection result. Since our method comprehensively considers the two kinds of temporal features (absolute positional information and relative change information) and effectively captures long-term dependencies in time series data, we can capture more complete and accurate temporal features to effectively improve the accuracy of depression detection.

## 4 EXPERIMENTS

In this section, we first show the details of the dataset and implementation. Then, we assess both the performance and efficiency of our proposed **FacialPulse** framework. Finally, ablation experiments are conducted to investigate the impact of the devised modules.

### 4.1 Experimental Setup and Details

*4.1.1 Datasets.* We evaluate the performance of our method on two depression datasets: the AVEC2014 dataset and an internally collected dataset. The AVEC2014 Depression dataset [34] consists of 300 videos from the 2014 Audio/Visual Emotion Challenge and Workshop, including "NorthWind" and "FreeForm" tasks. In the context of the "NorthWind" task, participants delve into a German fable entitled "Die Sonne und der Wind," where they read through its narrative. On the other hand, the "FreeForm" task demands not only answering a series of questions but also recounting a poignant childhood memory in the German language. Each task includes 150 video segments, with 80% of them allocated for training and the remainder for testing. In our experiments, we merge the samples from both tasks. Subsequently, we allocate 240 samples for training and 60 samples for testing. These videos are captured via webcams and microphones with an average duration of two minutes. Additionally, each video is labeled with the depression level, which is determined by the Beck Depression Inventory-II (BDI-II) questionnaire. Particularly, BDI-II is an estimation method of depression levels and has depression values ranging from 0 to 63, where 0-13 implies no depression, 14-19 mild depression, 20-28 moderate, and 29-63 severe depression.

The other internally collected dataset, named the Multimodal Dataset for Depression and Anxiety (MMDA) [17], was specifically designed for depression and anxiety detection. All participants are diagnosed by professional psychologists based on the combined Hamilton Rating Scale for Depression scores and Anxiety scores. MMDA includes visual, acoustic, and textual modalities, which are extracted from the original interview videos. In our experiments, we select all 300 depression detection video segments that are related to facial expressions from this dataset.

*4.1.2 Evaluation Metrics.* With the release of the AVEC2014 dataset, Root Mean Square Error (RMSE) and MAE are used as metrics for the 2014 Audio/Visual Emotion Challenge and Workshop. After that, these two metrics have been widely adopted to evaluate the performance of depression detection. For the sake of fairness, we also use RMSE and MAE as evaluation metrics in the experiments, which can be formulated as:

$$RMSE = \sqrt{\frac{1}{N} \sum_{i=1}^{N} (s_i - \hat{s}_i)^2}, \tag{11}$$

$$MAE = \frac{1}{N} \sum_{i=1}^{N} |s_i - \hat{s}_i|, \tag{12}$$

where $N$ is the number of participants, $s_i$ and $\hat{s}_i$ denote the true and predicted BDI-II scores for the $i$-th participant, respectively.

*4.1.3 Experimental Details.* During preprocessing, we utilize Dlib for face and landmark detection. As for ablation studies, OpenFace serves as an alternative detector. Each RNN in our dual-stream

network is bidirectional, which employs GRU with $k = 64$ output units for classification. A fully connected layer with a single unit is connected to the back of the RNN layer. We insert a dropout layer with a rate of 0.25 between the input and the RNN. Furthermore, three dropout layers with a rate of 0.5 are embedded in the remaining layers. The Adam optimizer with a learning rate of 0.001 is adopted. During classification, we choose the smooth $L_1$ Loss function, which is defined as follows:

$$loss = \begin{cases} 0.5(x)^2 & |x| < 1, \\ |x| - 0.5 & otherwise, \end{cases} \quad (13)$$

where $x_i$ represents the error between the predicted value and the true value. Compared to the Mean Squared Error, the smooth $L_1$ loss function appears to lower sensitivity to outliers, which can significantly boost the robustness against potential outliers in the data. The classification model is trained 500 epochs. All the experiments are conducted on a single RTX 3090 GPU with 24GB memory.

## 4.2 Performance Evaluation

*4.2.1 Comparison to Existing Approaches.* To verify the superiority of **FacialPulse**, we compare it with other state-of-the-art methods on the AVEC2014 dataset. Typically, methods based on deep neural networks present better performance compared to hand-crafted methods, which is primarily attributed to the fact that hand-crafted features rely on the expertise of researchers. In such cases, hand-crafted methods may not comprehensively mine depression cues, thereby decreasing prediction accuracy. As shown in Tab. 1, we report the results of comparative experiments with the evaluation metrics RMSE and MAE. Among all listed pioneer depression recognition methods, **FacialPulse** attains the top performance on MAE and second-best performance on RMSE. In particular, since the temporal features are deeply considered, **FacialPulse** surpasses with 1.5% RMSE improvements over the previous SOTA method [24] on AVEC 2014 datasets. By assessing RMSE and MAE, Fig. 6 (a) indicates that **FacialPulse** achieves the best overall performance among the three listed SOTA depression recognition methods.

**Table 1: Analysis of performance for different methods on AVEC2014 dataset, by evaluating RMSE and MAE.**

| Methods | RMSE | MAE |
|---|---|---|
| Baseline [34] /LGBP-TOP, SVR | 10.86 | 8.86 |
| Jan *et al.* [16] / EOH, LBP and LPQ, PLSR | 10.50 | 8.44 |
| Kaya *et al.* [19] /LGBP-TOP + LPQ | 10.27 | 8.20 |
| Zhu *et al.* [39] /Two CNN | 9.55 | 7.47 |
| Jazaery *et al.* [1] /Two C3D | 9.20 | 7.22 |
| Melo *et al.* [5] /Two C3D | 8.31 | 6.59 |
| Melo *et al.* [6] /ResNet-50 | 8.25 | 6.30 |
| Melo *et al.* [8] /Two ResNet-50 | 7.94 | 6.20 |
| Xu *et al.* [35] /MTB-DFE+SPG | 7.65 | 6.24 |
| Melo *et al.* [9] /MDN-152 | 7.65 | 6.06 |
| Niu *et al.* [24] /CNN+GCE+MSV | **7.56** | 6.01 |
| **FacialPulse** | 7.60 | **5.92** |

Furthermore, on an internally collected MMDA dataset, **FacialPulse** achieves a significant decrease in MAE compared to a baseline (SVM) (4.35 *vs* 3.87). These results demonstrate the significant competitiveness of the proposed **FacialPulse**, which can be attributed to the strong ability of our method to capture depression-related temporal features.

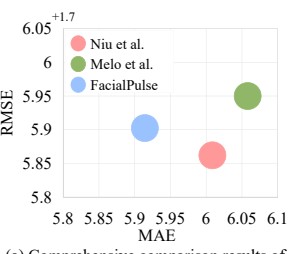 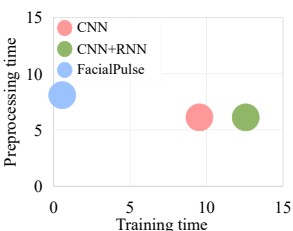

(a) Comprehensive comparison results of RMSE and MAE.

(b) Comprehensive comparison results of preprocessing time and training time.

**Figure 6: We conduct a comprehensive comparison of performance and experimental time. (a) presents the comprehensive RMSE and MAE compared with the leading three methods, and (b) represents the comprehensive preprocessing time and training time compared with two classical methods.**

*4.2.2 Computational Cost Evaluation.* Tab. 2 shows the experimental speed comparisons of the proposed approach and several representative baseline methods. All methods require similar preprocessing time and **FacialPulse** consumes two more hours than others due to more temporal information being considered. Noting that, due to the properties of parameter sharing and parallel computing in our method, the training time of **FacialPulse** is significantly less than that of others. To observe more intuitive results on preprocessing time and training time, Fig. 6 (b) shows that the proposed method is significantly closer to the zero point than others. The result clearly indicates that **FacialPulse** is significantly superior to other SOTA methods in terms of training speed.

**Table 2: Comparison of different methods on time cost including preprocessing and training.**

| Methods | Preprocessing | Training |
|---|---|---|
| CNN | 6h | 9.5h |
| CNN+RNN | 6h | 12.5h |
| **FacialPulse** | 8h | 0.5h |

**Table 3: Comparison of additional computational cost. "Param" denotes the parameterizable training size of the model. We also evaluate the GPU memory footprint.**

| Methods | Param | GPU |
|---|---|---|
| CNN | 11.6M | 6G |
| CNN+RNN | 24M | 9G |
| **FacialPulse** | 0.5M | 2.5G |

Additionally, Tab. 3 depicts the details of experimental costs including parameter sizes and GPU memory usage. Since **FacialPulse** has a small parameter space and employs a parallel computing strategy, it exhibits quite low training costs compared to others.

*4.2.3 The Impact of the Calibration Module.* To validate the effectiveness of the proposed calibration module, we conduct a confirmatory experiment. We first divide facial landmarks into seven regions. Then, different detectors (OpenFace and Dlib) are employed to detect landmark locations. Fig. 7 illustrates the mean distance between landmarks detected by different detectors. Using different detectors brings different noises and the calibration module aims to eliminate the noise and make them closer to the true position on the ground. Thus, this process can shorten the gap in the detection results of different landmark detectors.

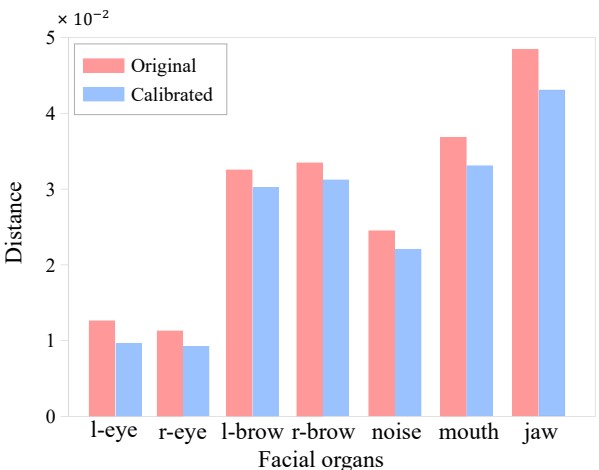

**Figure 7: Compare the average distance between different detected landmarks before and after using the calibration module. The abscissa represents the seven types of facial organs after grouping, and the ordinate represents the distance. The term "Original" represents the results obtained using the baseline method, while "Calibrated" denotes computation results after integrating our calibration module.**

From Fig. 7, we observe that after applying our calibration module, the detected differences of each organ significantly reduced and the average distance between the seven sets of landmarks decreased by 11%, which signifies an improvement in landmark detection accuracy. Due to the effectiveness of the proposed calibration module, we successfully eliminate noise and errors to obtain more accurate landmark positions.

## 4.3 Ablation Experiments

In this section, we explicitly investigate the influence of each module in the proposed framework **FacialPulse**, which provides evidence and a detailed explanation for the generated prominent results.

Tab. 4 shows the performance evaluated by RMSE and MAE under different ablation conditions. As a module is added, the values of RMSE and MAE decrease to a certain extent. Notably, in this

**Table 4: The impact of Kalman filter and the calibration strategy in the Facial Landmark Calibration Module (FLCM).**

| Methods | RMSE | MAE |
|---|---|---|
| Default | **7.60** | **5.92** |
| w/o Kalman Filter | 7.75 | 6.00 |
| w/o Calibration | 8.04 | 6.09 |

process, the Kalman filter effectively eliminates detection error and prediction error, while the optical flow prediction module effectively eliminates jitter noise. Each module in the Facial Landmark Calibration Module aims to obtain more accurate landmarks and further improve the accuracy of depression detection.

**Table 5: The impact of the two branches $(r_1 + r_2)$ in the Facial Motion Modeling Module (FMMM). $r_1$ denotes the modeling of absolute positional information, while $r_2$ denotes the modeling of relative change information.**

| Methods | RMSE | MAE |
|---|---|---|
| $(r_1 + r_2)$ | **7.60** | **5.92** |
| $r_1$ | 7.72 | 5.98 |
| $r_2$ | 8.00 | 6.07 |

In addition to performing ablation experiments on FLCM, we also study the impact of the two branches in the Facial Motion Modeling Module. Tab. 5 exhibits the depression detection results of each branch and combined branch. It can be clearly seen that the performance is significantly improved after integrating the two branches. By integrating absolute positional information and relative change information, the proposed method captures more comprehensive temporal features and achieves superior performance on both two metrics in facial depression detection tasks.

## 5 CONCLUSION

We propose a novel framework (**FacialPulse**) aimed at improving the accuracy and speed of depression recognition utilizing facial expressions. **FacialPulse** consists of two key modules: Facial Motion Modeling Module (FMMM) and Facial Landmark Calibration Module (FLCM). FMMM is designed to effectively capture temporal features by employing bidirectional processing and addressing long-term dependencies. Notably, FMMM's parallel processing capabilities and gate mechanism substantially accelerate training speed. Meanwhile, FLCM endeavors to reduce information redundancy by utilizing facial landmarks instead of original images, thereby enhancing recognition accuracy by eliminating errors associated with facial landmarks. Extensive experiments are conducted on the AVEC2014 and MMDA datasets, demonstrating the superior performance of **FacialPulse**. In future work, we aim to explore the integration of other complementary modalities into our proposed architecture to further enhance model performance.

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
