# OpenReview forum: "FacialPulse: An Efficient RNN-based Depression Detection via Temporal Facial Landmarks"
_acmmm.org/ACMMM/2024/Conference — MM2024 Oral_

### Official Review · Reviewer_XTCc · 2024-04-27

**Rating:** 4
**Confidence:** 2

**Summary:**

This paper integrates temporal facial landmark information to detect depression, which is novel. The facial landmarks are calibrated to achieve more accurate depression detection. However, the experiment is not very comprehensive.

**Strengths:**

1. Landmark calibration is useful and makes sense in this task. Designing a facial motion modeling module for depression detection is novel.
2. The GPU memory cost is reduced and training time is short.

**Limitations:**

1. Some prior knowledge about FACS and AU should be introduced to help the authors understand this work.
2. You want to use landmarks to obtain some action unit information. So why don't you contain the AU detection task? It may be helpful for the depression detection task.
3. It misses some important related work such as [1], and should be proved on more datasets such as AVEC2013.
[1]Pan Y, Shang Y, Shao Z, et al. Integrating deep facial priors into landmarks for privacy preserving multimodal depression recognition[J]. IEEE Transactions on Affective Computing, 2023.

**Suitability:**

3

---

### Official Review · Reviewer_bcDA · 2024-05-15

**Rating:** 6
**Confidence:** 4

**Summary:**

The paper employs a sophisticated approach to facial expression recognition in individuals with depression by utilizing RNN. In this study, the RNN model is applied to analyze facial expressions by not only considering the visible changes in facial features but also by incorporating detailed facial landmarks and the temporal sequence of these features. By tracking the movement and deformation of facial landmarks  over time, the model can capture subtle changes in expressions that are indicative of emotional states, particularly those associated with depression. The proposed method shows high accuracy and high recognition speed in 2 depression datasets.

**Strengths:**

The proposed method considered temporal features and performed well in depression detection for the micro-expression change of depression people. Also, facial landmark calibration and combination of forward and reverse information flows help to improve accuracy of the method.

**Limitations:**

One recommendation for enhancing the quality and comprehensiveness of the paper is to ensure that it is thoroughly examined for any missing references, particularly the mention of Taylor's Formula. Thoroughly examining the paper for the missing reference to Taylor's Formula, and any other key concepts, ensures that the paper meets high standards of academic completeness.

**Suitability:**

3

---

### Official Review · Reviewer_XJaq · 2024-05-31

**Rating:** 6
**Confidence:** 4

**Summary:**

It provides a depression detection technology by temporal facial landmarks. The idea is very novel and substantial.

**Strengths:**

It employs the temporal dynamics of facial expressions to extract more information than traditional static images and use face marks as input features to decrease data redundancies.

**Limitations:**

It is better to incorporate more advanced technologies.

**Suitability:**

3

---

### Meta-Review · Area_Chair_b1bg · 2024-07-10

**Recommendation:** Accept (Oral)
**Confidence:** 5

**Metareview:**

this paper investigates the problem of depression detection. by introducing an RNN-based architecture, FacialPulse extracts the temporal information from the video clip and achieves competitive performance over multiple benchmarks.

initially, the paper received ratings of A, A, and BA. the reviewers seem happy with the use of temporal information, despite concerns over the limited related work section. during the rebuttal, the authors' feedback has mostly addressed the concerns.

since the AC was not able to finish the meta review in-time, the PC stepped in and went through all the reviews and the rebuttal carefully. after consideration, the PC recommends Accept, and strongly encourages the authors to include more comparisons with the current work and potentially other attempts at incorporating the temporal information for similar tasks.